# Divided Village, Divided Identity? Exploring the Professional Identity of Teachers Amid the Geopolitical Configuration in Al-Ghajar

**DOI:** 10.3390/bs13110878

**Published:** 2023-10-24

**Authors:** Yonit Nissim, Eitan Simon

**Affiliations:** Department of Education and Learning, Tel Hai College, Qiryat Shemona 1220800, Israel; eitans@telhai.ac.il

**Keywords:** teachers, professional identity, geopolitical configuration, Ghajar

## Abstract

This quantitative research is based on a validated research questionnaire. It presents a preliminary exploratory study examining perspectives of professional identity as reflected in self-reports of the teachers of Al-Ghajar, a village on Israel’s northern border, given its unique geographical and geopolitical configuration. The current study is the first of its kind, clarifying the teachers’ perception of their professional identity in the unique space in which they live and work via a questionnaire completed by 61 teachers of both genders from kindergarten through high school. The findings show that there are strong positive correlations: “love for the profession” and “self-fulfillment” have a very strong positive correlation (0.831). There is also a positive correlation (0.430) between the indicators of “professional skills” and “professional unity” suggesting that the teachers’ perceptions of professional identity are holistic in all examined dimensions. These perceptions are motivated by a strong sense of mission that influences their profession and their sense of satisfaction. Al-Ghajar, as a unique enclave, has created a geopolitical configuration that contributes to the construction of the professional identity of teachers in the village. The research conclusion indicates that the ability to legitimize one’s role has important implications for the quality of teaching, as it can help teachers form familiar, affiliated, and secure identities. These are key traits since a positive sense of professional self is a prerequisite for job satisfaction and resilience.

## 1. Introduction

Researchers in the field of psychology and education recognize the importance of professional identity. This is a multifaceted construct that serves as a framework for introspective contemplation, encompassing both emotional and cognitive dimensions related to self-awareness and self-definition. A rich body of research dealt with the professional identity of teachers around the world. However, there is still a lack of dedicated studies that show the construction of teachers’ professional identity as rooted in the geopolitical space in which they live, study, and work, a space that cognitively affects and forms their professional identity. This study seeks to examine the professional identity of teachers in the village of Al-Ghajar in northern Israel, with its population of Alawite Muslims. Because the village’s boundaries to the north actually penetrate Lebanon, until July 2022, Ghajar remained a secluded village, with residents having the freedom to leave but strict restrictions on outsiders entering the village, enforced by the state. However, in the summer of 2022, the village was allowed to open its gates and welcome visitors and travelers.

The primary objective of this research initiative is to explore the perspectives of teachers residing in the village, based on the geopolitical configuration that has implications specifically in relation to shaping their professional identity, and the space in which they operate as an isolated minority between geographical, religious, and ethnic boundaries of integration and exclusion [1]. This multifaceted process is closely intertwined with emotional and cognitive responses, resulting in a nuanced and evolving interpretation of identities. It is important to emphasize that, as far as we are aware, there has been limited prior research focused on the village of Al-Ghajar. Most of the existing studies have primarily concentrated on political, geographical, and historical aspects. Consequently, this research aims to fill this significant gap by delving into the uncharted territory of Al-Ghajar educators’ professional identity, offering a deeper insight into the intricate connection between location, identity, and the teaching profession within the unique geopolitical context of the village [2,3,4].

## 2. Literature Review

### 2.1. Space and Cognition

The geographical location of one’s birthplace serves as a foundational anchor for a multifaceted array of cultural, value-laden, personal, and linguistic affiliations. This sense of belonging is deeply ingrained in the physical terrain, connecting individuals to a specific area of land [5]. It is imperative to underscore the need to expand the conceptualization of a “place” beyond its geographical coordinates. Such a view must be attuned to the particular contextual application within the realm of education and pedagogy.

Long-term patterns of behavior are often cognitive reflections of stable individual tendencies such as attitudes, values, and identity [6], i.e., an individual’s environmental identity based on their actions and behavior. Self-identity is influenced by social factors, where people use self-descriptions to show their affiliation or aspiration to belong to certain social groups [7]. A study in environmental social psychology has examined how the connections between social factors and a sense of place contribute to the development of place identity among individuals living in a specific geographic area [8].

Moreover, narratives of place-related personal experiences highlight the complex relationship between physical spaces and the diverse tapestry of human experiences, acting as a means for individuals to forge deep connections with their environment. A recent study suggests that the emotional responses elicited by a particular physical space vary significantly among individuals. This underscores the dynamic and highly personalized nature of the emotional landscape that unfolds within the context of specific spatial settings, further emphasizing the intricate interrelationship between individuals and their physical surroundings [9]. Distinct environments and the representations of space are subject to a variety of cognitive interpretations that, in turn, evoke disparate emotional experiences. This perceptual divergence gives rise to the individual construction of cognitive maps, each uniquely tailored by one’s accumulation of experiences and aspirations for future emotional states. These cognitive maps serve as intricate mental constructs that aid individuals in navigating and understanding their surroundings.

Furthermore, neurological investigations have revealed that an individual’s assessment of their environment prompts the activation of an internal model, intricately generated by their body to facilitate their interaction with the external world. This model encapsulates a repository of information concerning their emotional disposition, physiological condition, and aspirations for their future emotional and physical states. In essence, it represents a dynamic, personalized blueprint that informs and guides an individual’s behavior and perception within their surroundings. This neurological perspective underscores the profound interplay between the human mind, body, and the external environment, wherein the internal model plays a pivotal role in shaping an individual’s cognitive and emotional engagement with the world around them [10].

Human cognitive experiences encompass more than just physical locations; they include spaces within our memories and imagination. The representation of space in our minds is emotionally charged and can affect us even without a direct connection to the physical environment. This interplay between perception, emotion, and mental representations of space highlights the complex nature of human experience. Individuals can be deeply influenced by mental constructs of space, regardless of their immediate physical presence, evoking strong emotional responses [1,3].

**Figure 1 behavsci-13-00878-f001:**
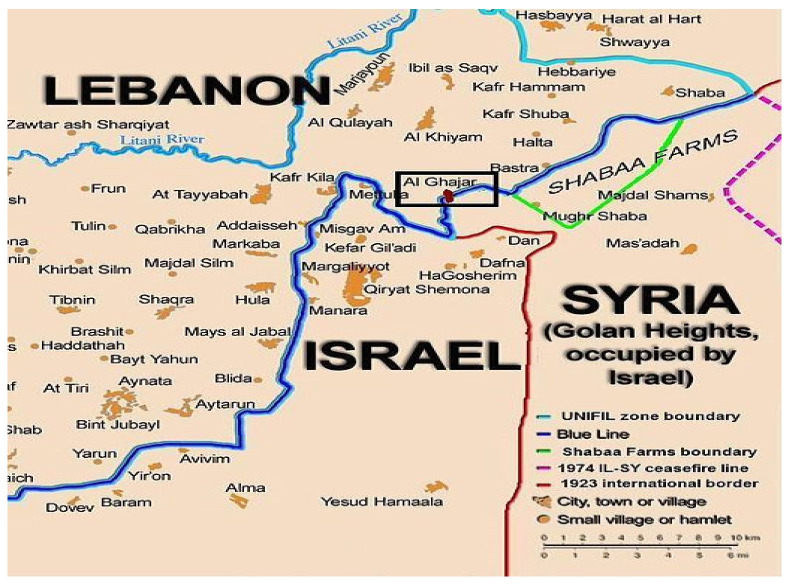
Location of Al-Ghajar (File: Ghajar highlighted.JPG–Wikimedia Commons, https://creativecommons.org/licenses/by-sa/2.5/deed.en (accessed on 29 August 2023)).

### 2.2. The Village of Al-Ghajar

As seen in the map above (Figure 1), Al-Ghajar, a village in the north of Israel, is an enclave situated at the crossroads of Israel’s geographic, political, and social borders, neighboring officially hostile countries. The majority of its population is Alawite Muslims. This situation dates back to the aftermath of the 1967 Six-Day War when the Golan Heights became part of Israel while its Syrian counterpart remained outside. Al-Ghajar, however, straddles both Lebanese and Israeli territories due to Lebanese authorities rejecting it, citing its Alawite residents’ Syrian ethnicity. In response, Al-Ghajar’s residents embraced Israeli identity, obtaining Israeli identification cards, acknowledging Israeli institutions, and following Israeli education directives. They adopted the Israeli Druze curriculum. This intricate geopolitical setup offers a unique case study, shedding light on identity, citizenship, and the complex interplay between ethnicity and political loyalties in a region fraught with geopolitical tensions [11].

The students of Al-Ghajar embarked on their educational journey by engaging with the curricula prescribed for the Arab and Druze sectors in Israel, which encompassed subjects including Hebrew language instruction. Additionally, the village teachers received their professional training through Israeli academic institutions. As previously elucidated, until the year 2022, Al-Ghajar retained a unique status as a quasi-enclave, characterized by stringent access restrictions. Residents of Israel were prohibited from entering the village, while residents were permitted to enter Israel for educational and employment opportunities, albeit under specific guidelines.

This historical overview underscores the significance of Al-Ghajar’s geographical location. It indicates that this spatial context not only shapes broader identities but also plays a substantial role in professional identity, especially among the village’s teachers. The intricate interplay between geopolitical boundaries, access constraints, educational frameworks, and individual experiences within this distinct space contributes to the unique professional identity of Al-Ghajar’s educators. This offers an important opportunity to explore the dynamic connection between space and identity, particularly in professional roles, where external constraints and personal agency together mold one’s professional self-concept [12].

### 2.3. Perceptions and Beliefs about the Role of the Teacher

Perceptions and cognitive images significantly influence various aspects of teaching, including methodologies, classroom behavior, student interactions, teacher development, and academic achievements. These interconnections ultimately affect teaching practices and outcomes, emphasizing the need for careful consideration in discussions about educational practices and policy development. Mendelsohn [13] has noted a significant correlation between the teachers’ self-image and professional application as reflected in their work in class [13].

Perceptions are shaped in the interaction between personal, professional, political and external contexts. Teachers’ perceptions are closely linked to the social and cultural practices in their particular environment. This context includes a wide range of experiences that go beyond their role as educators, involving life experiences and external influences. Additionally, their professional context, which is connected to, but separate from, their broader life experiences, involves the combination of theoretical knowledge and practical insights gained during their time as teachers [14]. Nevertheless, a comprehensive examination of these contexts is imperative, as it sheds light on their synergistic role in shaping professional perceptions and identities among teachers.

These perceptions and identities, in turn, serve as pivotal determinants of the quality of educational practice. Consequently, an exploration of these interdependencies may provide valuable insights into the intricate processes that underlie effective pedagogy and may underscore the critical importance of understanding the multifaceted nature of contextual influences in the realm of education [15]. Brown and colleagues [16] emphasize that perceptions consist of cognitive and emotional dimensions, which regulate perceptions that are sensitive to change. The cognitive dimension refers to the interaction between knowledge and individual beliefs about knowledge that influence the acceptance, adoption, and structuring of new knowledge. Changing perceptions is challenging.

In their research, Barselai et al. [17] delved into the intricate issue of the Druze community as a minority in the Golan Heights, specifically in the context of the significance attributed to higher education studies, both within Israel and overseas. Their conclusion affirms the unequivocal impact of geographical space on the formation of personal identity, which, in turn, exerts a discernible influence on the development of professional identity during preservice teacher training. This research also shed light on the existence of a complex interplay of emotions and perceptions among these teachers. Among the discerned dynamics, the study identified a prevailing sense of apprehension regarding potential alienation and discrimination faced by the Druze community. In juxtaposition to these concerns, there was also a sense of security and economic stability derived from their association with Israel, an element that might also substantially influence the identity formation of teachers in Al-Ghajar. These multifaceted perceptions and images engendered by their unique geopolitical context are not confined to mere cognitive constructs but have far-reaching implications [17].

### 2.4. Teacher’s Professional Identity

In this study, the term “identity” is used to conceptualize the complex nature of teacher identity, as presented in various perspectives, including Erikson’s theory of identity [2]. Personal identity is a subjective and intricate notion that revolves around maintaining consistency and coherence across various facets including personality, life experiences, and memories. It encompasses an individual’s journey through past, present, and future within the context of their professional development. It involves a profound awareness of one’s actions, existence, and the accumulation of specific experiences and perspectives within one’s chosen profession. Essentially, individuals are in a constant state of evolution within the professional realm as they navigate their career paths [18]. It is commonly acknowledged that the most significant phases of identity development tend to occur during youth [19].

Professional identity research encompasses the dynamic interplay between external contexts and internal personal processes. These factors collectively influence the thoughts, behaviors, and expressions of both novice and experienced educators, shaping how they perceive and engage with their personal experiences, values, beliefs, and expectations in the context of their teaching profession [20]. Kuzminsky [21] notes that a teacher’s professional identity is the answer to the question “Who or what am I, as a professional?” This fundamental question involves personality traits and values and is rooted in place and time.

Studies attempting to define the concept of “teachers’ professional identity” encounter a lack of clarity that is expressed in two possible approaches. The first seeks to present an objective definition of the term, while the second advocates subjective and individual interpretation [22]. Fitzgerald [23] notes six elements that define professional identity—basic training, professionalism and reliability, being informed, environmental activity, collaborations, and self-reflection. These elements are a link to the general group that upholds these norms, that is, the local community and society on the one hand and the professional community on the other. The social environment thus also shapes a teacher’s professional identity. Unlike “cross-border teachers” [24] who live and teach in different communities with diverse identities, the teachers of Al-Ghajar live and work in a closed, homogeneous geographical community. This is especially important when discussing closed communities.

There is a strong link between how teachers perceive their professional identity and the challenges they encounter in their role. In particular, teachers who have a strong sense of self-worth and a clear understanding of their professional identity tend to be more engaged and influential in their educational roles [25]. This heightened self-awareness not only empowers them but also contributes to their overall sense of fulfillment and positivity in their teaching endeavors [26]. Teachers’ awareness of their identity and its components is essential for their personal development. It strengthens their professional commitment and increases motivation and a sense of ability and efficacy [27]. Research has shown that teachers with a strong and stable teacher identity deal better with professional identity tensions [28] and can provide appropriate guidance in the socialization of their students. Furthermore, researchers suggest that teachers who have a holistic and solid identity demonstrate greater emotional involvement and enthusiasm at school [29]. Charles [30] suggests that teachers should look for purpose in their professional role as educators. Having a goal allows them to implement effective pedagogy, develop meaningful curricula, and connect relevant policies with clear objectives. This is essential for educators working outside their home context as it reinforces teachers’ pedagogy and identity as professionals [30].

### 2.5. Research Aim and Question

This preliminary quantitative study aims to explore the professional identity of teachers in the unique geographical setting of Al-Ghajar, which remains an unexplored topic in the literature. Hence, the research question is as follows: How do the teachers of Al-Ghajar perceive their professional identity in relation to the geopolitical–social–religious configuration in which they live?

## 3. Methodology

The current study is predominantly quantitative, investigating a phenomenon within its true context, especially when the boundaries between the phenomenon and the context are not entirely clear. In this regard, each teacher is considered as a unit of analysis. This undoubtedly influences and shapes personal and professional identity alike.

### 3.1. Participants

This work is based on a descriptive, cross-sectional study through a web-based survey conducted in March and April 2023. The survey questions were adapted and modified from a previously published questionnaire regarding teachers’ professional identity [22]. This study was conducted using a convenience sampling of the general teacher population. Participants were 61 teachers, of whom 14 (23.0%) were women and 47 (77.0%) were men, aged 24–60 (M = 42.66, SD = 8.91). About half (49.2%) of the respondents teach in elementary school, and the rest teach in high school (29.5%), middle school (13.1%), and kindergarten (6.6%). Most of the subjects are married (86.9%), and all are academics (100.0%). Table 1 below presents the distribution of respondents by demographic variables.

### 3.2. Procedure

The questionnaire was sent to a distribution list of all 70 teachers in Al-Ghajar via email. Of these, 61 teachers responded, providing a high response rate (87%). The questionnaire was open for data collection on Google Drive for three months. We sent email reminders to ensure a high return rate. The data were collected and underwent statistical tests as is customary, with an average (M) and standard deviation (SD) calculated for each statement. We used the Statistical Software Package for the Social Sciences, version 23 (SPSS Inc., Chicago, IL, USA).

### 3.3. Research Tool

This research was based on data collection, quantitative data processing, and analysis based on a validated research questionnaire (attitude survey) on the professional identity of teachers. The survey questionnaire presented to participants drew on a validated research tool that has been used and validated in several previous studies [22]. This research tool was adapted to the needs of the current study. We also added one open-ended qualitative question, the responses to which underwent content analysis using Atlas.ti software (ATLAS.ti Scientific Software Development GmbH, Berlin, Germany). It was validated (face validation) by four content experts, each with a PhD in education, in order for the quantitative research to be valid and reliable. We performed the usual statistical calculations: *T*-tests for independent samples and Pearson correlations.

The questionnaire employed in this study consisted of 41 statements rated on a 1–5 Likert scale and organized into five distinct themes. The first theme, “love for the profession”, encompassed 9 statements (items 1, 3, 5, 6, 10, 11, 19, 37, 38). The second theme, “sense of mission”, comprised 9 statements (items 2, 16, 17, 20, 21, 26, 27, 30, 32). The third theme, “professional skills”, included 12 statements (items 7, 8, 9, 18, 22, 25, 31, 33, 34, 36, 39, 40). The fourth theme, “the reputation of the profession”, featured 6 statements (items 4, 14, 23, 29, 35, 41). Finally, the fifth theme, “self-realization”, contained 5 statements (items 12, 13, 15, 24, 28). To account for negatively worded items (items 3, 7, 12, 25, 29, 32), the scales were reversed. For each theme, the participants’ average responses were calculated, thereby establishing five research indicators. Detailed characteristics of the measuring instrument are presented in Table 2 below.

The questionnaire concluded with one open-ended question: “How would you define your professional work in relation to the area in which you live?” The written answers underwent content analysis into categories (the data are presented in Section 4 following the quantitative data). Qualitative content analysis is non-linear and is characterized by de-contextualization and re-contextualization. It requires breaking the data into pieces [31] by dividing the original text into meaning units and condensing and coding those units [32].

The responses to the open-ended question addressed the characteristics of professional role perceptions as teachers in this particular village. The data collected underwent a five-stage open coding process. Each researcher read the answers several times individually and then together. The content analysis of the text and the conversion of the data into content categories enabled the comprehensive observation of the phenomenon under investigation [33]. Next, we created categories, while looking for common elements between the theory and our research material and coded recurring ideas and topics [34].

### 3.4. Ethics

All necessary steps were taken to ensure ethical standards were upheld. The college Ethics Review Board granted ethical approval. Participants were assured that their responses would be confidential and were reminded that their participation in the survey was voluntary. The questionnaire was anonymous and informed consent forms were signed. The questionnaire was distributed through Google Forms. At the beginning of the questionnaire, there was a personal address to the participants, guaranteeing the application of the rules of ethics, anonymity, and informed consent.

## 4. Findings

First, the prevalence and the center and distribution indices were calculated for demographic variables, as presented in Table 1. The five study indices were then calculated with their level of reliability via Cronbach’s alpha (Table 2). In addition, averages and standard deviations were calculated for all statements in the questionnaire (Table 3).

The Pearson correlations calculated for the five study indices are presented in Table 3. In addition, the correlations between the sense of mission and love for the profession, as well as between the sense of mission and self-fulfillment, are presented in Figure 2.

In order to examine the associations between the five research indices and demographic variables, *t*-tests for independent samples and Pearson correlation tests were performed (Table 5, Table 6, Table 7 and Table 8).

As mentioned, the questionnaire consisted of 41 statements on a Likert scale of 1–5 divided into five topics. The first topic, love of the profession, was addressed in nine statements. The second, a sense of mission, was also addressed in nine statements. The third, professional skills, was addressed in 12 statements. The fourth, reputation of the profession, was covered by six statements. The fifth, self-fulfillment, was covered by five statements. The remaining six statements were written in negative language and were therefore calculated with reversed scales. For each participant, an average was calculated for each topic, and thus, the five research indicators were defined. Table 2 presents the general characteristics of the research indicators.

**Table 2 behavsci-13-00878-t002:** General characteristics, averages, standard deviations, and reliability of research indicators (N = 61).

Indicator	No. of Statements	Min.	Max.	M	SD	α
Love of the profession	9	1.78	5	3.96	0.75	0.856
Sense of mission	9	2.33	5	4.02	0.65	0.808
Professional skills	12	2.08	5	4.24	0.48	0.814
Reputation of the profession	6	2.33	5	4.01	0.61	0.661
Self-fulfillment	5	2.4	5	3.89	0.67	0.666

The reliability of the indicators was found to be high according to Cronbach’s α, a statistic that characterizes a high degree of stability and consistency in the responses of the subjects in the statements of each indicator. Table 3 below presents the averages for each statement, divided by indicator.

**Table 3 behavsci-13-00878-t003:** Averages and standard deviations for the research statements (N = 61).

Love of the Profession	M	SD
I am sure I was right to choose to become a teacher	4.08	1.01
I don’t like being a teacher	2.70	1.65
I feel good about being in education	4.30	0.78
I am happy I chose teaching	3.97	1.06
I am drawn to being a teacher	4.02	1.12
I always wanted to be a teacher	3.75	1.14
I think that anyone who doesn’t like being a teacher shouldn’t be one	3.97	1.10
I am drawn to work in the field of education	4.10	0.94
I am satisfied to be working in education	4.15	0.83
Sense of mission	M	SD
I see my profession as a mission	4.33	0.83
Teaching plays a major role in my life	4.02	0.94
I don’t see myself leaving the field of education	3.85	1.18
I always thought that my mission is to be a teacher	3.82	1.06
People who don’t see teaching as a mission shouldn’t be teachers	4.13	1.02
I feel I am suited to being a teacher	4.48	0.81
I am at peace with my choice to teach	4.13	0.96
For me, teaching is a mission	4.13	0.94
I am not sure I will stay much longer in teaching	2.74	1.45
Professional skills	M	SD
I often have doubts whether I am suited to being a teacher	2.21	1.29
I have the personal ability to be a good teacher	4.56	0.70
I think I am a professional teacher	4.48	0.81
I think I have the professional skills to be a good teacher	4.52	0.65
I am sure I have the qualities to be a good teacher	4.48	0.62
I lack basic skills to be a teacher	1.57	0.94
I have the right approach to students	4.56	0.65
I am proficient in the secrets of the teaching profession	3.64	0.95
I am sure I have the competence to be a good teacher	4.51	0.65
I know what I have to do in teaching	4.54	0.57
I think only few teachers can define themselves as teaching professionals	3.44	1.09
The COVID-19 pandemic has made me more creative	3.93	0.85
Reputation of the profession	M	SD
When the media offends the status of teacher I take it personally	3.98	0.92
When someone is disdainful, I feel they are offending me	3.69	1.31
When I see a teacher, I hold them in high esteem	4.41	0.78
I am ashamed of being a teacher	1.33	0.72
If people think I have another job, I correct them and say I am a teacher	3.98	1.07
The COVID-19 pandemic has strengthened the status of teachers	3.34	1.11
Self-fulfillment	M	SD
I think I will actualize myself in a profession other than teaching	2.90	1.31
I think that teaching is the profession that suits me best	3.66	1.12
I think I am fulfilling myself in teaching	3.89	1.03
I can express who I am in teaching	4.20	0.93
I believe I will succeed in teaching	4.62	0.64

This table shows all the indicators and categories tested as they make up the professional identity of the research population. In the next stage, we checked the correlations between the different categories.

### 4.1. Correlations between the Research Indicators

Pearson tests were performed to examine the correlations among the research indicators. The results are shown in Table 4 below.

As shown, there are positive associations of medium–strong intensity between all research indicators. In other words, an increase in the levels of each of the indicators will also raise the levels of the other indicators.

We used a Pearson test to clarify the behavioral model from the above indicators and examined the correlations and effect that the sense of mission has on two variables: the love of the teaching profession and the sense of fulfillment (professional satisfaction). The results are presented below. Figure 2 demonstrates the correlation between the sense of mission and love of the profession, as well as between the sense of mission and self-fulfillment: the greater the sense of mission, the greater the love of the profession and the greater the self-fulfillment. A significant correlation was found, indicating that mission consciousness was a significant element in shaping and consolidating the professional identity of the village teachers. This contributes to the formation of a more “complete” professional identity.

#### 4.1.1. Education

To examine the differences in research indicators between bachelor’s degree and master’s degree holders, *T*-tests were performed for independent samples. Table 5 below shows the averages for both groups and the test results.

As shown, the average of bachelor’s degree holders is slightly higher than the average of master’s degree holders in most indicators (except for profession reputation), although not significantly. In other words, there are no differences according to education.

#### 4.1.2. Gender

To examine the differences in the research indicators between men and women, *T*-tests were performed for independent samples. Table 6 below shows the averages for both groups and the test results.

As shown, women’s averages are higher than men’s averages in most indicators (except for profession reputation). In the self-fulfillment indicator, the differences are significant t(59) = 3.38, *p* < 0.01, so we can say that the women feel greater self-fulfillment than the men.

#### 4.1.3. Age, Teaching Seniority, and Number of Children

Pearson tests were conducted to examine the correlations between age, teaching seniority, and number of children and the research indicators. The results are shown in Table 7 below.

Most of the correlations were found to be weak and insignificant. However, a positive correlation of moderate and significant intensity (r = 0.355, *p* < 0.01) was found between profession reputation and age, so that the older the teacher, the stronger the profession reputation is felt.

#### 4.1.4. Family Status

To examine the differences in the research indicators between single and married individuals, *t*-tests were performed for independent samples. Table 8 below shows the averages for both groups and the test results.

As shown, there are no significant differences between the averages for single and married teachers for any of the indicators.

There are strong positive correlations (marked in green) between several categories; for example, “love for the profession” and “self-fulfillment” have a very strong positive correlation (0.831 **), suggesting that teachers who have a high level of love for their profession tend to experience a high level of self-fulfillment. There is also a positive correlation (0.430 **) between the indicators of “professional skills” and “professional unity”, suggesting that teachers who report higher professional skills tend to also report a stronger sense of professional unity. Similarly, “sense of mission” and “professional skills” have a strong positive correlation (0.654 **), implying that teachers who feel a strong sense of mission in their work also tend to report higher professional skills.

#### 4.1.5. Qualitative Findings

This section is based on content analysis into categories of the open-ended question: “How would you define your professional work in relation to the area in which you live?” The responses addressed the characteristics of the respondents’ professional role perceptions as teachers in this particular village.

Of the 61 teachers who responded to the questionnaire, 35 (57%) chose to answer this question. Four teachers (11%) chose to use the word “mission”, seven chose to use the word “excellent”, and nine (25%) chose to use the word “good”.

We used Atlas.ti software version 5 to conduct the analyses. Twelve teachers (30%) referred to concepts of wanting to help educate the next generation of village residents, while mentioning social–ideological motives, using terms such as “minority”, “helping my village”, and “sacred work”, in order to maintain social cohesion within the village, thereby preserving the unique character of the village residents in order to ensure its continued existence.

Some quotes from the answers in the teachers’ own words are as follows:

“Teaching is a mission for us, as members of a minority. We have a responsibility to safeguard our children’s future.”

“Teaching is a challenging and beautiful endeavor, a sacred calling that demands considerable effort and unwavering dedication.”

“For me, teaching is a mission dedicated to advancing our profession, which, in my eyes, encompasses my entire village.”

“I hold deep appreciation for my students, with whom I share the same village, and I wholeheartedly give of myself in their education.”

When asked to define their professional role in their village, all respondents used positive terms such as “excellent”, “good”, “mission”, and “desire to help”. This suggests a strong commitment among this social minority to uphold social cohesion as an essential aspect of their professional educational work. They put significant effort into incorporating this commitment into their educational roles, seeing it as integral to their personal identity, often referring to Al-Ghajar as “my village” and emphasizing their identity as a minority within the community.

## 5. Discussion and Conclusions

This study examined how the distinctive context of Al-Ghajar village influences the professional identity perceptions of its teachers. The primary question was as follows: How do these teachers perceive their professional identity given their unique geopolitical, social, and religious environment? The findings revealed a nuanced and intricate professional identity influenced by various factors. Notably, their strong sense of mission played a pivotal role in shaping and reinforcing this professional identity, providing a sense of purpose and self-worth while defining their contributions to society. This sense of mission was intricately tied to their passion for the profession and their personal satisfaction. We find a very strong positive correlation between their “love for the profession” and “self-fulfillment”. Between the “sense of mission” and “professional skills”, we find a strong positive correlation implying that teachers who strongly identify with a sense of mission in their work tend to report higher levels of professional skills.

Moreover, the teachers’ responses to the open-ended question underscored their unwavering commitment as a social minority to uphold social cohesion through their educational endeavors. They consider their work an integral aspect of their identity, often stating, “This is my village”, and they emphasize their minority status. Importantly, they consistently used positive terms like “excellent” and “good” to describe their work, reflecting their genuine desire to contribute to the education of the village’s future generations. Their responses also revealed social and ideological motivations, particularly the preservation of their community’s distinct character, which they view as a sacred and indispensable mission for the village’s enduring existence. This underscores their dedication to social sustainability and continuity. The teachers’ perceptions reflect context-dependent social and cultural practices, and the context includes experiences prior to and outside the teaching experiences. As an exceptional enclave village, Al-Ghajar has forged a unique geopolitical configuration that significantly contributes to the construction of these teachers’ professional identities.

The teachers of Al-Ghajar operate as an isolated minority between geographical, religious, and ethnic boundaries of integration and exclusion [35]. The geographical configuration serves as a general space for personal professional conduct. In our opinion, it is a significant factor in shaping this identity, which is motivated by a sense of personal, professional, and communal mission of a minority, rooted in a complex geopolitical enclave.

In the professional context, these findings can refer to theoretical and practical experiences while teaching. These perceptions are in line with several studies [35,36] that suggested that teachers tend to evolve from the interaction between personal, professional, political, and external contexts. A well-developed professional identity could also improve teachers’ confidence in their decision to work in education, as well as their commitment to the profession. It is hard to separate the contexts. However, it is important to examine them in order to understand affinities as an integral part of shaping professional perceptions and identities as predictors of quality educational practice.

The findings in this paper demonstrate that the professional identity of the teachers of Al-Ghajar is well established, which indicates a high commitment to persevere [36]. Their self-perception and perspective may be an authentic expression that directly explains their professional identity. The current research indicates that the ability to legitimize one’s role has important implications for the quality of teaching, as it can help teachers form familiar, affiliated, and secure identities, which are all key traits, since a positive sense of professional self is a prerequisite for their job satisfaction and resilience [37].

The findings of this study are also consistent with earlier studies [38,39] that demonstrate how professional identity is shaped based on the characteristics of general identity, and how teachers “perceive” their place in society and its institutions and play a significant role in it while legitimizing their positions [39]. Teachers who are a national religious minority tend to preserve their community and sense of mission [40]. A fairly recent study revealed characteristics of tradition, integration, and participation among teachers in an Australian school, characterizing their essence as a “shared sacred mission” focused on the concepts of community, faith, life, and community [41].

Identity is shaped by culture and involves both individual cognition and social processes. Identity formation is not just personal; it is also a political process where individuals may choose to align with or challenge dominant societal norms and expectations [42,43]. Studies dealing with teacher professional identity have made it clear that the characteristics of general identity (gender, nationality, and race) impact the construction of professional identity and that they are intertwined [42].

Dealing with the issue of professional identity in teaching in the context of physical space is of great importance. Barselai et al. (2022) [17] suggested that the geographical space also influences the development of professional identity in preservice teacher training. Al-Ghajar, as a unique enclave, has created a geopolitical configuration that contributes to the construction of the professional identity of teachers in the village. Thus, their professional identity is a reflection of the place where they live and teach.

These conclusions provide valuable insights into the complex interplay of factors that shape teachers’ professional identities and the connections between various dimensions of their professional experiences. This study sought to unveil the perceptions of the professional identity of the teachers in Al-Ghajar, revealing strong connections between emotions, thoughts, and professional experiences. This comprehensive identity, referred to as a holistic professional identity, encompasses all facets of professional, personal, and regional elements. It includes a passion for the profession, a sense of purpose, professional skills, reputation, and self-fulfillment, all woven into a cohesive concept deeply rooted in the local community and society just like a beautiful mosaic with strong threads of emotion, thought, cognition, and experience connecting everything, forming a quilt that is a cherished part of their local community and society. We believe that this research has broad implications dealing with the relationship between professional identity and geographic space—a matter worth deepening the research on.

### 5.1. Recommendations for Future Research

This is a preliminary exploratory study. We believe that there is room to expand the research and examine in depth the types of motives, factors, and components that characterize and structure this identity, in an attempt to further understand its correlations and connections. It would be appropriate to conduct follow-up studies with in-depth interviews.

### 5.2. Research Limitations

The small size of the sample (61 teachers) might be seen as a limitation. However, since the entire possible population for this particular study was only 70, the high response rate (87.14%) makes it a suitably representative sample.

## Figures and Tables

**Figure 2 behavsci-13-00878-f002:**
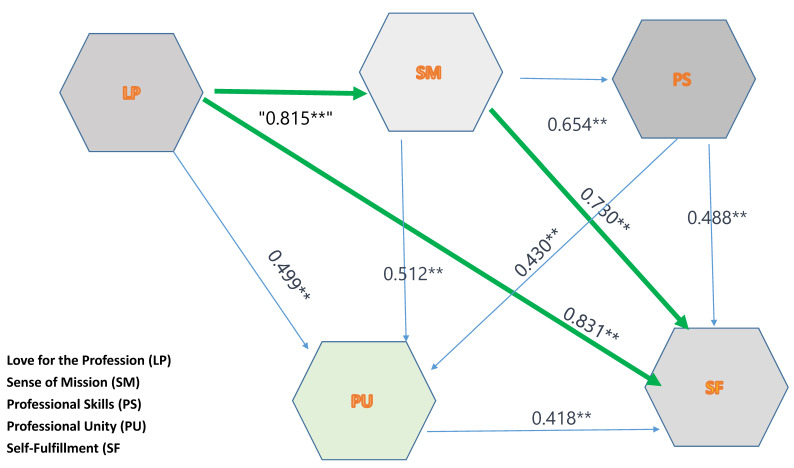
Correlations between the different variables. (Note: ** - statistical finding).

**Table 1 behavsci-13-00878-t001:** Distribution of demographic variables among all research participants (N = 61).

	N	%	Min.	Max.	M	SD
Gender						
Male	47	77.0				
Female	14	23.0				
Age			24	60	42.66	8.91
Family status						
Single	8	13.1				
Married	53	86.9				
No. of children			0	6	3.13	1.40
Years of teaching seniority			1	42	16.92	11.05
School type						
Kindergarten	4	6.6				
Elementary	30	49.2				
Middle	8	13.1				
High	18	29.5				
Other	1	1.6				
Education						
B.A./B.Sc./B.Ed.	25	41.0				
M.A./M.Sc./M.Ed.	36	59.0				

**Table 4 behavsci-13-00878-t004:** Pearson coefficients among the research indicators (N = 61).

Indicators	1	2	3	4
1. Love of the profession	--			
2. Sense of mission	0.815 **	--		
3. Professional skills	0.542 **	0.654 **	--	
4. Profession reputation	0.499 **	0.512 **	430 **	--
5. Self-fulfillment	0.831 **	0.730 **	488 **	418 **

** *p* < 0.01.

**Table 5 behavsci-13-00878-t005:** Research indices—bachelor’s and master’s degrees and *t*-test results (N = 61).

	Bachelor’s Degree (N = 25)		Master’s Degree (N = 36)		
Indicator	M	SD	M	SD	t
Love of the profession	4.02	0.74	3.91	0.76	0.56
Sense of mission	4.17	0.62	3.91	0.66	1.59
Professional skills	4.28	0.46	4.21	0.50	0.51
Profession reputation	3.99	0.63	4.03	0.61	0.21
Self-fulfillment	4.00	0.70	3.82	0.65	1.04

**Table 6 behavsci-13-00878-t006:** Research indices—men and women and *t*-test results (N = 61).

	Men (N = 47)		Women (N = 14)		
Indicator	M	SD	M	SD	t
Love of the profession	3.86	0.73	4.28	0.72	1.86
Sense of mission	3.94	0.65	4.26	0.62	1.63
Professional skills	4.23	0.48	4.28	0.51	0.36
Profession reputation	4.02	0.61	4.00	0.64	0.09
Self-fulfillment	3.74	0.62	4.39	0.62	3.38 **

** *p* < 0.01.

**Table 7 behavsci-13-00878-t007:** Correlations between age, teaching seniority, and number of children and the research indicators (N = 61).

Indicator	Age	Seniority	No. of Children
Love of the profession	0.135	−0.036	−0.027
Sense of mission	0.055	−0.108	−0.005
Professional skills	0.080	0.012	0.253
Profession reputation	0.355 **	0.236	0.168
Self-fulfillment	0.077	−0.039	0.047

** *p* < 0.01.

**Table 8 behavsci-13-00878-t008:** Research indices—single and married and *t*-test results (N = 61).

	Single (N = 53)		Married (N = 8)		
Indicator	M	SD	M	SD	t
Love of the profession	4.31	0.79	3.91	0.73	1.43
Sense of mission	4.38	0.62	3.96	0.64	1.70
Professional skills	4.21	0.43	4.24	0.49	0.19
Profession reputation	3.79	0.75	4.05	0.59	1.10
Self-fulfillment	4.28	0.85	3.83	0.63	1.75

## Data Availability

The data presented in this study are available on request from the corresponding author.

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
