# Peer review of "Divided Village, Divided Identity? Exploring the Professional Identity of Teachers Amid the Geopolitical Configuration in Al-Ghajar"

_behavsci, 2023, doi:10.3390/bs13110878_

Round 1

Reviewer 1 Report

Thank you for the opportunity to review this article. This is a rather well-structured article which can benefit the relevant field when published. I have a few concerns however, that is detailed here:

1. The authors need to explain efforts done to ensure this study is appropriate ethically. Ethical clearance details must also be discussed.

2. How did the authors select and approach participants? What did the authors do to avoid recruitment bias?

3. Too many short paragraphs (less than six sentences).

4. Contextual analysis in discussion is lacking, considering that the context of this research is one of its plus points.

Author Response

Leather for the reviewer

Article: behavsci-261147

Title:  a divided village – a complete professional identity? Geo-political configuration in the identity of teachers in the village of Ghajar.

Dear reviewer,

First, we would like to thank you wholeheartedly for your eye-opening comments. These have been tremendously helpful in guiding the major changes we have made to this paper. We would like to emphasize the following points that have been substantially modified in line with your comments

All your comments have been taken into account and the article has been modified accordingly.  We rewrote the following chapters: the introduction, the summary, the discussion and the conclusions. We added, expanded and deepened the important points you commented on. We were based on new studies published recently.

I will answer each comment separately. For convenience, I have attached your comments) in a bold font( and answered below each comment in accordance with what appears in the corrections made in the body of the article.

  • The authors need to explain efforts done to ensure this study is appropriate ethically. Ethical clearance details must also be discussed.

The questionnaire was distributed through Google Forms.

We have added explanations about maintaining the rules of ethics

Before the beginning of answering the questionnaire, there was a personal appeal to the participants in the following formulation

Dear teacher

We are asking for your help in filling out the questionnaire dealing with the professional identity of the teacher workers in Rager village.

We will thank you if you answer in the affirmative and fill out the questionnaire which will help us continue to encourage people to engage in teaching as an important profession for the future development of the village

If you do not want to answer the questionnaire, you can leave the link

The questionnaire is anonymous without any ability to identify the respondents

The duration of the questionnaire is about 7 minutes

Thanks the authors

  1. How did the authors select and approach participants? What did the authors do to avoid recruitment bias?

The email with the questionnaire was sent to all 100 teachers who teach and live in the village according to a distribution list. The questionnaire was anonymous and the answers were collected in Google Drive.

  1. 3. Too many short paragraphs (less than six sentences).

Most of the manuscript parts were rewritten. Paragraphs have been reorganized and modified.

  1. Contextual analysis in discussion is lacking, considering that the context of this research is one of its plus points.

This part was rewritten

Thank you very much

The authors

Reviewer 2 Report

Thank you for the opportunity to review your manuscript.

Introduction - generally sets the scene well. May be a bit of repetition e.g. the paragraph at the top of p,2 seems to repeat points already made on p.1.

Does the phrases “a lack of dedicated case studies that show the construction of teachers' professional identity as rooted in the geopolitical space in which they live, study, and work” draw a conclusion, which the research question set out to answer?

I wasn’t quite sure about the teachers in this village. Do they live and work there? Or do they live elsewhere and travel in? Does the reader need to know more about the numbers of teachers and schools in the village, and their type? Ah – I have now come to p.4 where you mention that the teachers live and work in the community. It seems misplaced there. Could that come earlier? Similarly on p.5 you mention some of the different types of school that might be useful earlier to ‘paint the picture’.

Literature review

I wonder if paragraph 3 about the Six-Day War should go into the Introduction.

Again, repetition about the study’s focus “on the teacher community of the divided village of Ghajar, whose geopolitical characteristics constitute a highly unusual geographical location”.

The paragraph about the definition of a case study seems misplaced, as it interrupts the discussion about different senses/meanings of ‘place’.

I think the general line of your argument in section 2.1 is that perceptions and beliefs about the role of the teacher are shaped by a myriad of experiences, influences, and emotions. These influence personal and professional identities (leading to Section 2.2). Not surprisingly perhaps, many of those who are new to teaching have not yet developed a fuller sense of professional identity, or possibly feel uncomfortable with it – see p.4. This is to be expected as they are only at the beginning of becoming part of that professional group.

After then mentioning that the teachers live in this closed community, you come to mention study by Keskin and a study by Charles. These two paragraphs seem oddly placed.

Section 2.3. Should the content here come earlier? It restates what you have already mentioned.

Methodology

I am not an expert in statistical methodology, so apologies if the points I raise are irrelevant. I just ask some questions which occur to me.

What does it been to say the questionnaire “has been validated by four content experts with a PhD in education”?

Why did you feel it necessary to collect data on marital status and number of children? I see you come to this on Table 6, but it made no sense to me.

What ethical protocols were followed? What permissions were required and obtained? Did participants have the right to withdraw. Was anonymity guaranteed? Was confidentiality assured? How did you know that you had responses form 61 different individuals (as opposed to some responding more than once?)? Were there any conflicts of interest? Would/could participants have felt obliged to answer in certain ways? Consider: if they know that the results are going to be used in your work and published, with such a small sample size, their identities could be at risk of being revealed once results are looked at (as there are only 4 teachers in Kindergarten, and 8 in Middle Schools for example).

As I say, I am no expert in statistics and so I found the results in the Tables rather confusing. On the Likert scale, was 1 low and 5 high? Or were they mixed?  Again, due to my own lack of expertise in statistics, I could not understand Table 3 or Figure 1.

I can make more sense of statements such as “a positive correlation of moderate and significant intensity … was found between profession reputation and age, so that the older the teacher, the stronger the profession reputation is felt” although this is not surprising. Similarly, I found the qualitative findings more accessible.

Does the reader need to be able to see a copy of the questionnaire?

Top of p.10: “which is a small village in northern Israel with only 2800 residents, located on and across the Lebanese border” – should this come in the Introduction?

Discussion and conclusions

Why use bold type for certain phrases in this section?

Strangely enough, I am not getting a sense of the participants coming across in your discussion and conclusions. It seems that many of them feel that they are working in an unusual environment, and this may have developed a strong sense of professional identity amongst them. You don’t pick out feelings of tension or of conflict, or of particular issues they face living and working in this village. Maybe the fact that they are also residents of the village is significant.

A few presentational issues:

Look to eliminate repetition.

Try to bring related points together more.

Font sizes seem to be vary within and between some paragraphs. For example, the first paragraph of the Introduction has font of different sizes within it.

Punctuation errors e.g. p.6: “The remaining six statements. Were written in negative language and therefore were calculated with reversed scales”.

I have made some suggestions for some re-ordering and editing of the material, and also to look out for typos.

Author Response

Leather for the reviewer

Article: behavsci-261147

Title:  a divided village – a complete professional identity? Geo-political configuration in the identity of teachers in the village of Ghajar.

Dear reviewer,

First, we would like to thank you wholeheartedly for your eye-opening comments. These have been tremendously helpful in guiding the major changes we have made to this paper. We would like to emphasize the following points that have been substantially modified in line with your comments

All your comments have been taken into account and the article has been modified accordingly.  We rewrote the following chapters: the introduction, the summary, the discussion and the conclusions. We added, expanded and deepened the important points you commented on. We were based on new studies published recently.

I will answer each comment separately. For convenience, I have attached your comments) in a bold font( and answered below each comment in accordance with what appears in the corrections made in the body of the article.

  1. How did the authors select and approach participants? What did the authors do to avoid recruitment bias?

The email with the questionnaire was sent to all 100 teachers who teach and live in the village according to a distribution list. The questionnaire was anonymous and the answers were collected in Google Drive.

We have added explanations about maintaining the rules of ethics

  1. Too many short paragraphs (less than six sentences).

Most of the parts of the manuscript were rewritten. Paragraphs have been reorganized and modified.

  1. Contextual analysis in discussion is lacking, considering that the context of this research is one of its plus points.

This part was rewritten as well.

5 etc. –All your important comments have been addressed in all the rewritten parts.

  1. I have made some suggestions for some re-ordering and editing of the material, and to look out for typos.

Thank you very much. Your suggestions were welcomed and in the rewritten parts, we based our writing on them.

Thank you for your enlightening review

The authors

Reviewer 3 Report

Dear Authors,

Please find below suggestions to improve the manuscript.

Abstract:

    • As a quantitative study, what tools or instruments were used to gather data? Providing this will establish methodological credibility. The Introduction mentions a case study research design. Please align the information on your study design.
    • Add information about the sample.
    • The term "holistic in all examined dimensions" is somewhat ambiguous.
    • Avoid making general statements. If there are any specific statistics or notable trends that emerged from the data, consider highlighting one or two.
    • The relationship between Ghajar's unique status and the teachers' professional identity is intriguing. If possible, elucidate this connection further, even if briefly.
    • Consider stating the broader implications of the findings. How can this research benefit the field of education or inform policy in regions with similar geopolitical contexts?
    • The use of terms like "first of its kind" may come off as too bold unless supported with comprehensive literature review evidence. Ensure such claims are well-grounded. Moreover, the Introduction mentions: `Only few studies have dealt with the village of Ghajar, almost all of them focusing on political, geographical, and historical aspects.’
    • Avoid repetition, such as the use of "unique" multiple times.

Introduction

The uniqueness of Ghajar is mentioned multiple times. Consider consolidating this information to avoid repetition.

Discuss its geographical significance in one section, then its historical/social significance in another to ensure clarity.

The use of quotation marks around terms like "divided", "closed", "isolated", and "space" can be confusing. Determine if these are necessary or if there are clearer terms that can be used without quotes. The narrative about the village opening in the summer of 2022 and the subsequent visit by the college team seems fragmented. Consider reorganizing this chronologically for clarity.

Emphasize the importance of the study more directly. Why is it crucial to understand the teachers' identity in this specific context? How does it contribute to the larger conversation about teacher identities in complex geopolitical scenarios?

It's mentioned that few studies have addressed Ghajar and that many have tackled teacher identity worldwide, but the gap between them isn't clearly delineated. Highlight the uniqueness of studying teacher identity in the Ghajar context more distinctly.

The claim that the study is the "first of its kind" is made multiple times. Once should suffice, if supported with a comprehensive review of literature.

The research question is introduced, but the specific question itself isn't clearly articulated. Consider providing the direct research question for clarity.

Some transitions between paragraphs or ideas can be smoother. For instance, after mentioning the college's visit, the transition to the arising questions could be made more seamless.

The ending should succinctly summarize the importance, uniqueness, and objective of the study. Consider restructuring the last paragraph to serve this purpose more effectively.

This section is oddly formatted. Some characters appear to be smaller than others. Please check for consistency and compliance with the journal’s recommendations.

Literature review

  • The transition between the broad concept of geographical location and the specific case of Ghajar might be clearer if more direct links were provided.
  • Consider providing a bridge statement that directly ties the general concepts of geographical location to the specific context of Ghajar.
  • The paper touches on several different aspects of geographical location and professional identity, but there might be merit in providing more depth to each point, especially the neurological and emotional aspects of people's connection to space.
  • Consider elaborating more on the historical context of Ghajar to provide readers unfamiliar with the region a deeper understanding.
  • It's mentioned that Ghajar was captured during the Six-Day War in 1967 and has remained under Israeli control. However, it's also noted that a part of Ghajar remains under Lebanese rule. For clarity, you might want to explain how and why this division took place.
  • When introducing terms like 'professional identity', provide a clear and concise definition at the outset. This will help readers unfamiliar with educational or sociological jargon. It is stated that ‘The concept of professional identity refers to the professional choice of the individual, attitudes in relation to the profession chosen, the operation of professional decisionmaking and the sense of professional infrastructure. ‘, but it appears after 2 other pragraphs discussing professional identitity.
  • The mention of "cross-border teachers" and teachers in closed communities introduces an intriguing comparative element. Consider elaborating more on how the situation in Ghajar contrasts with these other contexts.
  • Be mindful of reiterating points. For example, the uniqueness of the study in relation to Ghajar and its teachers' professional identities is mentioned multiple times. While it's crucial to emphasize the novelty of the research, avoid being overly repetitive.
  • There are minor grammatical inconsistencies, such as "a single unit learning process that can be an individual, event, pprogram, sschool, or community." Ensure thorough proofreading to correct such typographical errors.

Methodology

Be specific about which versions of SPSS and Amos were used.

Mention the limitations of the snowball sampling method, like potential selection bias, and how it might affect the results. Did the initial teacher who distributed the questionnaire have any biases or personal relations that might influence the sampling?

If any follow-up was done to achieve the 87% response rate, it should be mentioned (e.g., reminders to non-respondents).

Mention which specific statistical tests were performed. This can help readers gauge the appropriateness and rigor of the analysis.

Elaborate on the content analysis process of the open-ended question. How were the categories derived? Was any specific software used for this analysis?

Findings

The findings section should be a separate one and not be comprised in the Methodology.

‘The five study indices were then calculated with their level of reliability via Cronbach’s alpha (Table 2.1). In addition, averages and standard deviations were calculated for all statements in the questionnaire (Table 2.2).’ - The five dimensions or factors. Have you conducted any CFA to see how the 41 variables load onto the five factors?

‘The Pearson correlations calculated for the five study indices are presented in Table 3. In addition, the correlations between the sense of mission and love for the profession, as well as between the sense of mission and self-fulfillment were presented using Amos software.’ – please revise the use of indices.

‘In order to examine the associations between the five research indices and demographic variables, t-tests for independent samples and Pearson correlations were performed (Tables 4-7).’ – Is the use of parametric tests appropriate in the case of your data? Please check the conditions and revise accordingly.

Then, you mention the correlations among ‘research indicators’. Please use the terminology consistently.

‘As shown, there are positive associations of medium-strong intensity between all research indicators.’ – Please mention all the three characteristics of a correlation and add the statistical significance.

Figure 1: In AMOS graphics, observed variables are represented as rectangles (or sometimes squares). They are often labeled with the actual variable names from your dataset.

Latent variables are depicted as ovals (or circles) in AMOS.

Arrows (pathways) run from these latent variables to the observed variables they are supposed to measure. This denotes that the latent variable is manifesting through the observed variables.

Love for profession, self-fulfilment and sense of mission are not directly observed variables and they should not be represented as rectangles. If you want to use Amos, please insert and discuss the path diagram.

Discussion

Some parts of the discussion seem to jump between concepts without clear transitions. When discussing holistic correlations between the various components of professional identity, concrete examples from the findings would strengthen the discussion. Ensure that the citation style is consistent throughout the section. Some references are numeric (e.g., "37 38"), while others are author-year (e.g., "Sultmann and Brown42"). This inconsistency can be confusing to readers. The theme of "holistic identity" is repeated multiple times. While emphasis is important, it might be more effective to succinctly state the point and then provide supporting evidence or insights without repeating the same phrases.

Comments regarding the language and the overall style are included in the text above.

Author Response

Leather for the reviewer

Article: behavsci-261147

Title:  a divided village – a complete professional identity? Geo-political configuration in the identity of teachers in the village of Ghajar.

Dear reviewer,

First, we would like to thank you wholeheartedly for your eye-opening comments. These have been tremendously helpful in guiding the major changes we have made to this paper. We would like to emphasize the following points that have been substantially modified in line with your comments

All your comments have been taken into account and the article has been modified accordingly.  We rewrote the following chapters: the introduction, the summary, the discussion and the conclusions. We added, expanded and deepened the important points you commented on. We were based on new studies published recently.

I will answer each comment separately. For convenience, I have attached your comments) in a bold font( and answered below each comment in accordance with what appears in the corrections made in the body of the article.

  1. How did the authors select and approach participants? What did the authors do to avoid recruitment bias?

The email with the questionnaire was sent to all 100 teachers who teach and live in the village according to a distribution list. The questionnaire was anonymous and the answers were collected in Google Drive.

We have added explanations about maintaining the rules of ethics

  1. Too many short paragraphs (less than six sentences).

Most of the parts of the manuscript were rewritten. Paragraphs have been reorganized and modified.

  1. Contextual analysis in discussion is lacking, considering that the context of this research is one of its plus points.

This part was rewritten as well.

5 etc. –All your important comments have been addressed in all the rewritten parts.

  1. I have made some suggestions for some re-ordering and editing of the material, and to look out for typos.

Thank you very much. Your suggestions were welcomed and in the rewritten parts, we based our writing on them.

Does the reader need to be able to see a copy of the questionnaire?

Depending on the journal's requirements, if it requests, it can be incorporated as an appendix at the end of the article

Top of p.10: “which is a small village in northern Israel with only 2800 residents, located on and across the Lebanese border” – should this come in the Introduction?

Discussion and conclusions

As mentioned, all the parts: introduction, summary, discussion and conclusions were completely rewritten based on the comments.

Why use bold type for certain phrases in this section?

The highlight in bold has been removed.

Strangely enough, I am not getting a sense of the participants coming across in your discussion and conclusions. It seems that many of them feel that they are working in an unusual environment, and this may have developed a strong sense of professional identity amongst them. You don’t pick out feelings of tension or of conflict, or of particular issues they face living and working in this village. Maybe the fact that they are also residents of the village is significant.

The points you raised are important and the introduction has been rewritten highlighting the fact that they are both residents of the village who were born and raised in it and also teachers who teach in this area.

A few presentational issues:

Look to eliminate repetition.

Try to bring related points together more.

Font sizes seem to be vary within and between some paragraphs. For example, the first paragraph of the Introduction has font of different sizes within it.

Punctuation errors e.g. p.6: “The remaining six statements. Were written in negative language and therefore were calculated with reversed scales”.

Comments on the Quality of English Language

The points you raised are significant. They have been modified in the manuscript in accordance with your coments.

Thank you for your enlightening review

The authors

Round 2

Reviewer 1 Report

Thank you authors for revising this manuscript. I think the manuscript is ready to be published.

Author Response

thank you very  much

Reviewer 2 Report

Thank you for making some change to your paper. I think these have improved it, although I think some of my comments on the previous version have not been fully addressed.

Language in section 2.1 is quite expansive.

In one place, you use the term conduits twice on three lines – would be better style to change one of them.

You also use the term locales a lot.

Section 2.2

Can you be clearer about the location/status of Ghajar. It is not clear. You write that “Ghajar assumes the unique status of a divided village, nestled within the triangulated nexus of Israel's geographical, political, and social borders”, but then you go on to write about “its counterpart” - what is that? Then you say “Ghajar, however, finds itself situated within the confines of Lebanese territory”, which seems to contradict what you said previously.

Do you mean that it found itself in Lebanese territory back in 1967? If so, how did it become part of Israel? You mention its quasi-enclave status until 2022. What happened then?

Later on, you write that the village is located on and across the Lebanese border.

It might be that all the information is here, but it is not coming across clearly.

Section 2.3

At the end of the first sentence, you have a note 1. Where is this?

You return to discuss teacher identity. Are you in danger of repeating points already made?

Section 2.4 on professional identity

In terms of language, you tend to use rather convoluted phrasing. An example is "Identity, within the realm of sociological and psychological discourse, constitutes a multifaceted construct that encapsulates an individual's self-perception. This self-perception is composed of an amalgamation of perceptions and beliefs pertaining to the self, which are systematically organized within the cognitive framework of a schema”.

You are making the same points again and again. It is very straightforward to note that teacher identity and professional identity are complex notions. By the time we get to the methodology we are on p.7 of 16.

Methodology.

You still have the sentence that I queried previously, namely: the questionnaire “has been validated by four content experts with a PhD in education”.

On a minor point, this should read each with a PhD in education. But, more substantively, what does it mean to say they had validated it?

I am not a statistician and so cannot comment on the use of statistics.

Can you be sure that no one completed the questionnaire more than once?

Section 4: Discussion and conclusions

You write "The findings indicate that the teachers in the village Ghajar operate as isolated minorities between geographical, religious and ethnic boundaries of integration and exclusion [36]".

Why is this referenced to another 1986 study? What in your findings supports this statement?

As before, I am not getting much of a sense of the participants coming across in your discussion and conclusions, and the feelings of tension or of conflict they experience, or of particular issues they face living and working in this village.

I hope these comments will be helpful.

There are a number of errors that need attention. I highlight examples here:

“Abstract :T”:

“T The”

“as reflected in a self-reported of the”

“summer of 2022 the village opened his gate to visitors” – the village is not a ‘his’

“teachers that were born and rise” – should be born and raised

“The space in which they operate as isolated minorities between geographical, religious and ethnic boundaries of integration and exclusion “ – doesn’t make sense.

through emails a total of 70.

An anonymous questionnaire and informed consent.

All the necessary steps were taken to ensure compliance with the rules of ethics, an anonymous questionnaire and informed consent.

In the introduction to the questionnaire, we addressed the research population as follows: The questionnaire was distributed through Google Forms .At the beginning of the questionnaire, there was a personal appeal to the participants, which guarantees the rules of ethics, anonymity and informed consent.

The Teachers' perceptions

consistent with previous studies [38,39]. That demonstrate how professional

the Love of the profession

A space that has a constructive effect on the teachers' professional identity.

Sideheading Reference should be References

Inconsistent paragraph spacing throughout.

Reference style is inconsistent (e.g. order of author surnames and forenames; italicisation).

Author Response

8.10.23

Leather for the reviewer

Article: behavsci-261147

Title:  "Divided Village, Divided Identity? Exploring the Professional Identity of Teachers amidst the Geo-political Configuration in Ghajar

Dear reviewer,

First, we would like to thank you wholeheartedly for your eye-opening comments. These have been tremendously helpful in guiding the major changes we have made to this paper. We have made significant changes based on all your comments as you can see in "Track the changes". We would like to emphasize the following points that have been substantially modified in line with your comments.

All your comments have been taken into account and the article has been modified accordingly.  We rewrote the all the following chapters: the introduction, the summary, the discussion and the conclusions. We added, expanded and deepened the important points you commented on. We were based on new studies published recently.

I will answer each comment separately. For convenience, I have attached your comments) in a bold font( and answered ,below in red ,each comment in accordance with what appears in the corrections made in the body of the article.

Thank you for making some change to your paper. I think these have improved it, although I think some of my comments on the previous version have not been fully addressed.

Thank you very much. We continue to improve it according to your comments.

Language in section 2.1 is quite expansive.

In one place, you use the term conduits twice on three lines – would be better style to change one of them.

 We have changed one of the term  to means.

You also use the term locales a lot.

We chanced it and new it appears only once

Section 2.2

Can you be clearer about the location/status of Ghajar. It is not clear.

We added a map that illustrate the geo graphical place.

You write that “Ghajar assumes the unique status of a divided village, nestled within the triangulated nexus of Israel's geographical, political, and social borders”, but then you go on to write about “its counterpart” - what is that? Then you say “Ghajar, however, finds itself situated within the confines of Lebanese territory”, which seems to contradict what you said previously.

Do you mean that it found itself in Lebanese territory back in 1967? If so, how did it become part of Israel? You mention its quasi-enclave status until 2022. What happened then?

We clarified this and add a paragraph as follows:

Ghajar, however, finds itself divided within the confines of Lebanese territory and Israel’s, albeit encountering a peculiar set of circumstances. The Lebanese authorities adamantly rejected the incorporation of Ghajar into their territorial jurisdiction, contending that the Alawite ethnicity of its inhabitants naturally aligned them with Syrian citizenship. Consequently, the residents of Ghajar found themselves in a unique predicament, where the Lebanese government refused to extend citizenship rights to them due to their Alawite background. Since 1967, the village has been divided into two parts between two deferent states, Lebanon and Israel. That they do not establish good neighborly relations

Later on, you write that the village is located on and across the Lebanese border.

It might be that all the information is here, but it is not coming across clearly.

We clarified it and add a map We have marked the village with a circle to better visually clarify its unique location. We belleve that now it is easier to understand the complex situation.

Section 2.3

At the end of the first sentence, you have a note 1. Where is this?

We corrected it. It was written by mistake.

You return to discuss teacher identity. Are you in danger of repeating points already made?

We rewrite this part and corrected it .

Section 2.4 on professional identity

In terms of language, you tend to use rather convoluted phrasing. An example is "Identity, within the realm of sociological and psychological discourse, constitutes a multifaceted construct that encapsulates an individual's self-perception. This self-perception is composed of an amalgamation of perceptions and beliefs pertaining to the self, which are systematically organized within the cognitive framework of a schema”.

You are making the same points again and again. It is very straightforward to note that teacher identity and professional identity are complex notions. By the time we get to the methodology we are on p.7 of 16.

We rewrite and shortened this section.  

Methodology.

You still have the sentence that I queried previously, namely: the questionnaire “has been validated by four content experts with a PhD in education”.

On a minor point, this should read each with a PhD in education. But, more substantively, what does it mean to say they had validated it?

Werewrite this part and  corrected this. In this type of research, it is important to perform face validation by content experts in the research filled. This is the usually procedure common in a quantitative research (for validation and reliability).

I am not a statistician and so cannot comment on the use of statistics.

We added more detailed explanations about the questionnaire and the statistical tests performed

Can you be sure that no one completed the questionnaire more than once?

The findings were collected in Google Drive. The statistical tests take into account standard deviation, and sampling errors of this type as in quantitative research. Therefore reliability tests and statistical validation are performed.

Section 4: Discussion and conclusions

You write "The findings indicate that the teachers in the village Ghajar operate as isolated minorities between geographical, religious and ethnic boundaries of integration and exclusion [36]".

Why is this referenced to another 1986 study? What in your findings supports this statement?

As before, I am not getting much of a sense of the participants coming across in your discussion and conclusions, and the feelings of tension or of conflict they experience, or of particular issues they face living and working in this village.

We corrected it rewrite this part and addd some of the quotes from the teachers in their own words:

One teacher expressed, "Teaching is a mission for us, as members of a minority. We have a responsibility to safeguard our children's future." Another teacher reflected, "Teaching is a challenging and beautiful endeavor, a sacred calling that demands considerable effort and unwavering dedication." Yet another teacher emphasized, "For me, teaching is a mission dedicated to advancing our profession, which, in my eyes, encompasses my entire village."  Another teacher articulated, "I hold deep appreciation for my students, with whom I share the same village, and I wholeheartedly give of myself in their education."

I hope these comments will be helpful.

Your review was of great value for us

Thank you very much

The authors

Reviewer 3 Report

Dear Authors,

Please find below some suggestions for your paper.

The title seems to be a bit unclear and fragmented. It appears to be touching upon several different topics: division in a village, professional identity, geopolitical configuration, and the identity of teachers in Ghajar village. It's important for a title to clearly and concisely represent the content of the paper. It might be helpful to simplify and streamline the title for clarity.

Here’s a possible revision for consideration:

"Divided Village, Divided Identity: Exploring the Professional Identity of Teachers Amidst the Geo-political Configuration in Ghajar"

Abstract:

The abstract has some redundancy and typographical errors, which should be corrected to enhance readability and professionalism. For example, the phrase “T The question arises” contains a typo, and the final sentence is somewhat repetitive of previous statements.

Please provide a brief description of the quantitative methods used to gather and analyze the data.

Introduction

Overall, the introduction provides an extensive overview of the research topic, background information, and the study's aims. It does well in emphasizing the unique contribution of the research to the existing body of knowledge.

Here are some suggestions:

·       Consider revising for brevity and clarity. Avoid long, complex sentences that could confuse the reader.

·       Ensure the text is thoroughly proofread to correct any grammatical errors or typos.

·       Clearly emphasize the unique contribution of the study to the field to underscore its importance and relevance.

Literature review

2.1 Space and cognition

Please find below some suggestions:

  1. Opt for concise and clear language to convey concepts more accessibly. Avoid overuse of complex and lengthy sentences.
  2. Trim repetitive or overly verbose sections to maintain reader engagement and ensure a concise presentation.
  3. Elaborate on the “recent study” mentioned for a more comprehensive understanding. Detailing key findings and methodologies would add depth to the discussion.
  4. Offer more insights into the neurological investigations, delving deeper into the mechanisms that shape individuals' assessments of their environments.

2.2 The village of Al-Ghajar (Ghajar)

·       Consider incorporating maps or diagrams to visually represent the geopolitical situation of Ghajar, aiding in better understanding.

·       Continuously highlight the relevance of the presented background information to the professional identity of teachers to keep the reader’s focus on the research question.

2.3. Perceptions and beliefs about the role of the teacher

Strengthen the link between the general discussion of teachers' perceptions and the specific context of Ghajar to make the relevance even more apparent to readers.

Please consider to shorten the literature review.

Methods

Further elaboration on how the research tool was adapted for the study would be beneficial. Providing insights into the modifications made can lend additional context and clarity to the methodology.

Further elaboration on how the research tool was adapted for the study would be beneficial. Providing insights into the modifications made can lend additional context and clarity to the methodology.

Offering a bit more detail on the specific statistical analyses and why they were chosen can enhance the reader's understanding of the methodology.

Discussing potential limitations of the methodology, including the reliance on self-reported data and the limitations of the questionnaire, can enhance the section’s thoroughness and transparency.

Providing more information on how content analysis was conducted for the open-ended question responses would lend greater insight into the qualitative aspect of the research.

Usually, the Methods section includes: Participants, Measures, Procedure, Data Analysis. Please consider to reorganize the Methods section.

The analysis in AMOS is flawed.

As I said, the variables in Figure 1 are latent variables. In Amos, as in other software, we do not use rectangles to represent LVs. Please test the factorability of the data.

As for the model, it would be extremely useful for it to be diagnosed not only globally (by fit indices), but also analytically (convergent and discriminant validity). To this end, it would be very useful to calculate such indicators as: Average Variance Extracted, Composite Reliability, Maximum Shared Variance, Average Shared Variance, or Heterotrait-Monotrait Ratio of Correlations (HTMT) (Henseler, J., Ringle, CM, & Sarstedt, M. (2015).

Please revise this section, replace the path diagram (those are not correlations).

Thank you!

Please check for clarity, typos, and verbosity.

Author Response

8.10.23

Leather for the reviewer

Article: behavsci-261147

Title:  "Divided Village, Divided Identity? Exploring the Professional Identity of Teachers amidst the Geo-political Configuration in Ghajar

Dear reviewer,

First, we would like to thank you wholeheartedly for your eye-opening comments. These have been tremendously helpful in guiding the major changes we have made to this paper. We have made significant changes based on all your comments as you can see in "Track the changes". We would like to emphasize the following points that have been substantially modified in line with your comments.

All your comments have been taken into account and the article has been modified accordingly.  We rewrote the all the following chapters: the introduction, the summary, the discussion and the conclusions. We added, expanded and deepened the important points you commented on. We were based on new studies published recently.

I will answer each comment separately. For convenience, I have attached your comments) in a bold font( and answered ,below in red ,each comment in accordance with what appears in the corrections made in the body of the article.

Please find below some suggestions for your paper.

The title seems to be a bit unclear and fragmented. It appears to be touching upon several different topics: division in a village, professional identity, geopolitical configuration, and the identity of teachers in Ghajar village. It's important for a title to clearly and concisely represent the content of the paper. It might be helpful to simplify and streamline the title for clarity.

Here’s a possible revision for consideration:

"Divided Village, Divided Identity: Exploring the Professional Identity of Teachers Amidst the Geo-political Configuration in Ghajar"

Thank you very much, we gladly accepted your suggestion and it was adopted as the title of the article.

Abstract:

The abstract has some redundancy and typographical errors, which should be corrected to enhance readability and professionalism. For example, the phrase “T The question arises” contains a typo, and the final sentence is somewhat repetitive of previous statements.

We corrected that. Thank you.

Please provide a brief description of the quantitative methods used to gather and analyze the data.

We rewrite and organize this part. We add detailed description of the methods qualitative and quantitative We also added several examples From the original quotes from the teachers answers to the open end question in their own words

Introduction

Overall, the introduction provides an extensive overview of the research topic, background information, and the study's aims. It does well in emphasizing the unique contribution of the research to the existing body of knowledge.

Here are some suggestions:

  • Consider revising for brevity and clarity. Avoid long, complex sentences that could confuse the reader.
  • Ensure the text is thoroughly proofread to correct any grammatical errors or typos.
  • Clearly emphasize the unique contribution of the study to the field to underscore its importance and relevance.

Literature review

2.1 Space and cognition

Please find below some suggestions:

  1. Opt for concise and clear language to convey concepts more accessibly. Avoid overuse of complex and lengthy sentences.
  2. Trim repetitive or overly verbose sections to maintain reader engagement and ensure a concise presentation.
  3. Elaborate on the “recent study” mentioned for a more comprehensive understanding. Detailing key findings and methodologies would add depth to the discussion.
  4. Offer more insights into the neurological investigations, delving deeper into the mechanisms that shape individuals' assessments of their environments.

We accepted your suggestions and rewrite most of the paper accordingly.

2.2 The village of Al-Ghajar (Ghajar)

  • Consider incorporating maps or diagrams to visually represent the geopolitical situation of Ghajar, aiding in better understanding.

We add a map and we have circled the unique location of the village on the map

  • Continuously highlight the relevance of the presented background information to the professional identity of teachers to keep the reader’s focus on the research question.

2.3. Perceptions and beliefs about the role of the teacher

Strengthen the link between the general discussion of teachers' perceptions and the specific context of Ghajar to make the relevance even more apparent to readers.

 We accepted your suggestions and rewrite most of the paper accordingly.

Please consider to shorten the literature review.

Methods

Further elaboration on how the research tool was adapted for the study would be beneficial. Providing insights into the modifications made can lend additional context and clarity to the methodology.

We have rewritten this part extensively.

Further elaboration on how the research tool was adapted for the study would be beneficial. Providing insights into the modifications made can lend additional context and clarity to the methodology.

Offering a bit more detail on the specific statistical analyses and why they were chosen can enhance the reader's understanding of the methodology.

Discussing potential limitations of the methodology, including the reliance on self-reported data and the limitations of the questionnaire, can enhance the section’s thoroughness and transparency.Providing more information on how content analysis was conducted for the open-ended question responses would lend greater insight into the qualitative aspect of the research.

Usually, the Methods section includes: Participants, Measures, Procedure, Data Analysis. Please consider to reorganize the Methods section.

The analysis in AMOS is flawed.

As I said, the variables in Figure 1 are latent variables. In Amos, as in other software, we do not use rectangles to represent LVs. Please test the factorability of the data.

 The methodology section was rewritten according to your suggestions, regarding  AMOS It was a Outdated software of AMOS . we desided to present the numerical data In a graphic model describing the relationship between the various variables and the statistical results. see Figure 1: Correlations between the different variables.

Your review was of a great value for us

Thank you very much

The authors

Round 3

Reviewer 2 Report

Thank you for the further changes. I think these have given a sharper focus to your work.

Further proof reading is essential. There are still many errors.

For example, taking the following:

"Narratives of Personal experiences related to places highlight the complex relationship between physical spaces and the diverse tapestry of human experiences. Acting as a means for individuals to forge deep connections with their environments."

This could be better expressed thus:

Narratives of personal experiences related to places highlight the complex relationship between physical spaces and the diverse tapestry of human experiences, acting as a means for individuals to forge deep connections with their environments.

Another example:

"Research’s shows that teachers with a strong and stable teacher identity deal better with professional identity tensions" should be:

Research shows that teachers with a strong and stable teacher identity deal better with professional identity tensions.

"partisipents" should be participants.

"Figuer 1" should be Figure 1

"teacher’s professional identity" (p.10) in this case should be teachers' professional identity (as you are writing about the professional identity of more than one teacher).

The title in the proof version has a very tiny question mark after the word Identity. If the question mark is retained, it clearly needs to be the same font size as the rest of the text in the title. 

Divided Village, Divided Identity? Exploring the Professional Identity of Teachers amidst the Geo-political Configuration in Ghajar

These are just examples. A thorough proof reading of the whole document is recommended.

The map is useful. Its source needs acknowledging  and you will need to confirm that you have permission to use the map (for copyright reasons). I did wonder if it has become a little squashed vertically. It would be worth checking.

As I have mentioned before, I am not a statistician and so cannot comment helpfully on your use of statistics. 

I have made some comments in my suggestions about the standard of English. There are still a lot of errors and I have given some examples of the sorts of issues I have noticed.

Author Response

Article: behavsci-261147

Title:  "Divided Village, Divided Identity? Exploring the Professional Identity of Teachers amidst the Geo-political Configuration in Ghajar

Dear reviewer,

First, we would like to thank you wholeheartedly for your eye-opening comments. These have been tremendously helpful in guiding the major changes we have made to this paper. We have made significant changes based on all your comments as you can see in "Track the changes". We would like to emphasize the following points that have been substantially modified in line with your comments.

All your comments have been taken into account and the article has been modified accordingly.  We rewrote the all the following chapters: the introduction, the summary, the discussion and the conclusions. We added, expanded and deepened the important points you commented on. We were based on new studies published recently.

I will answer each comment separately. For convenience, I have attached your comments and answered ,(in bold below) to each comment in accordance with what appears in the corrections made in the body of the article.

Further proof reading is essential. There are still many errors.

The paper has been carefully proofread and edited.

The map is useful. Its source needs acknowledging  and you will need to confirm that you have permission to use the map (for copyright reasons). I did wonder if it has become a little squashed vertically. It would be worth checking. 

This has been sourced and checked for copyright a link has been aded to the map.

Your review was of a great value for us

Thank you very much

The authors

Reviewer 3 Report

Dear Authors,

Thank you for all the efforts to improve the quality of the manuscript. There are visible improvements. More is to be done, though.

Abstract: When drafting an abstract, it could be useful to consult the APA JARS standards

https://apastyle.apa.org/jars

The provided abstract does not fully comply with the JARS standards as it lacks essential details about the research methodology, specific findings, and conclusions with implications.

While the abstract briefly mentions that it is a "quantitative research" and a "preliminary exploratory case study," it lacks critical details required by JARS standards, such as the research design (e.g., experimental or observational), sample size, materials used, outcome measures, and data-gathering procedures.

The abstract reports some general findings, such as the holistic nature of teachers' perceptions of professional identity and the influence of Ghajar's geopolitical configuration. However, it does not provide any specific information about effect sizes, confidence intervals, statistical significance levels, or any quantitative data that would be expected according to JARS standards.

Please report estimates related to the reliability of measures (e.g., internal consistency coefficients) in the method section.

Data analysis and analytic strategies lack.

"Indicator" and "index" are not the same things; they have distinct meanings in various contexts, especially in data analysis. The authors used them interchangeably and confusingly. Please correct this:

‘In order to examine the associations between the five research indices and demographic variables, t-tests for independent samples and Pearson correlations were performed (Tables 4-7).’

Have you conducted an exploratory factor analysis to identify underlying latent factors or constructs?

Please add a note to Figure 1 to explain its source.

In the Findings section it is mentioned that `We used Atlas software version 5 to do the analyses’. Please mention that in the Methods section as well. Atlas.ti is the name of the software.

The manner in which you employed the Atlas.ti software and the resulting outcomes are not clear.

Why the `summery’ at the end of the Findings? Please correct the typographical error.

Please check for grammar errors.

‘In the professional context, the finding can refers to theoretical and practical experiences while teaching [36]. These perceptions are in line with several studies that suggested that teacher tend to evolve from the interaction between personal, professional, political and external contexts. a developed professional identity could also improve teachers' confidence in their decision to work in education, as well as their commitment to the profession. It is hard to separate the contexts. However, it is important to examining them in order to understand affinities as an integral part of shaping professional perceptions and identities as predictors of quality educational practice [37].’

The text above says that the results are in line with several other studies, but none is cited.

Please check for grammar errors: ‘The finding in this paper demonstrate that the professional identity of the teachers of Ghajar is well established which indicates a high commitment to persevere [38]. ‘ It is not usually to say ‘the finding’.

‘A current research indicates that the ability to legitimize one’s role has important implications for the quality of teaching, as it can help teachers form familiar, affiliated, and secure identities, which are all key traits, since a positive sense of professional self is a prerequisite for their job satisfaction and resilience’ – Perhaps a recent research indicated …

‘These findings are also consistent with studies [40,41].’ – Which findings? Please rephrase the sentence. Something is lacking.

`Teachers who are a national -religious minority tend to preserve their community and sense of mission. a current study revealed characteristics of tradition, integration and participation of teachers in an Australian schools characterized their essence as a "shared sacred mission". Focused on the concepts of community, faith, life and community. [43]’ – Please replace current.

Please check for grammar errors: ‘Barselai, et al (2022) [2245] suggested that the geographical space influence on the development of professional identity during the training process of educators.’

Please check for grammar errors, improve flow and rephrase verbose statements.

Author Response

Dear reviewer,

First, we would like to thank you wholeheartedly for your eye-opening comments. These have been tremendously helpful in guiding the major changes we have made to this paper. We have made significant changes based on all your comments as you can see in "Track the changes". We would like to emphasize the following points that have been substantially modified in line with your comments.

All your comments have been taken into account and the article has been modified accordingly.  your coments are in blue and our answer are below. 

Thank you for all the efforts to improve the quality of the manuscript. There are visible improvements. More is to be done, though.

Abstract: When drafting an abstract, it could be useful to consult the APA JARS standards

https://apastyle.apa.org/jars

The provided abstract does not fully comply with the JARS standards as it lacks essential details about the research methodology, specific findings, and conclusions with implications.

This was add

While the abstract briefly mentions that it is a "quantitative research" and a "preliminary exploratory case study," it lacks critical details required by JARS standards, such as the research design (e.g., experimental or observational), sample size, materials used, outcome measures, and data-gathering procedures.

These have been added.

The abstract reports some general findings, such as the holistic nature of teachers' perceptions of professional identity and the influence of Ghajar's geopolitical configuration. However, it does not provide any specific information about effect sizes, confidence intervals, statistical significance levels, or any quantitative data that would be expected according to JARS standards.

This was add and corrected according to your comments

Please report estimates related to the reliability of measures (e.g., internal consistency coefficients) in the method section.

These appear in the findings.

Data analysis and analytic strategies lack.

These have been added.

"Indicator" and "index" are not the same things; they have distinct meanings in various contexts, especially in data analysis. The authors used them interchangeably and confusingly. Please correct this:

‘In order to examine the associations between the five research indices and demographic variables, t-tests for independent samples and Pearson correlations were performed (Tables 4-7).’ Indicator is the term used throughout.

Have you conducted an exploratory factor analysis to identify underlying latent factors or constructs?

Please add a note to Figure 1 to explain its source.

In the Findings section it is mentioned that `We used Atlas software version 5 to do the analyses’. Please mention that in the Methods section as well. Atlas.ti is the name of the software. Added and corrected

The manner in which you employed the Atlas.ti software and the resulting outcomes are not clear.

 All added

Why the `summery’ at the end of the Findings? Please correct the typographical error.

The summary information has been moved to the Discussion.

Please check for grammar errors. Proofread and edited

The manuscript was proofread and re- edited .

Your review was of a great value for us

Thank you very much

The authors
